# Analytical Investigation of Replica-Molding-Enabled Nanopatterned Tribocharging Process on Soft-Material Surfaces

**DOI:** 10.3390/mi15030417

**Published:** 2024-03-21

**Authors:** In Ho Cho, Myung Gi Ji, Jaeyoun Kim

**Affiliations:** 1Department of Civil, Construction, and Environmental Engineering, Iowa State University, Ames, IA 50011, USA; 2Department of Electrical and Computer Engineering, Iowa State University, Ames, IA 50011, USA; mji90@iastate.edu

**Keywords:** triboelectricity, tribocharge, contact electrification, elastomer, replica molding, PDMS

## Abstract

Nanopatterned tribocharge can be generated on the surface of elastomers through their replica molding with nanotextured molds. Despite its vast application potential, the physical conditions enabling the phenomenon have not been clarified in the framework of analytical mechanics. Here, we explain the final tribocharge pattern by separately applying two models, namely cohesive zone failure and cumulative fracture energy, as a function of the mold nanotexture’s aspect ratio. These models deepen our understanding of the triboelectrification phenomenon.

## 1. Introduction

The phenomenon of nanopatterned tribocharge formation [1] is receiving growing attention. Of special interest is its realization on the surface of soft, elastic materials for applications like energy harvesting [2] and flexible electronics. Recently, we developed a new, replica-molding-based technique to induce nanopatterned tribocharge on the surface of poly(dimethylsiloxane) (PDMS), a widely adopted elastomer [3,4]. It is based on our observation that the surface of PDMS becomes decorated with nanopatterned electric charge after being replica molded from nanotextured polymer molds [3].

The resulting tribocharge’s distribution patterns, imaged by Kelvin probe force microscopy (KPFM) [4] and electrohydrodynamic lithography (EHDL) [3], exhibited close correlations with the nanotexture patterns on the polymer molds, allowing the technique to be used to generate nanopatterned surface charge in a highly controlled fashion. Long-term monitoring of the tribocharge on the PDMS surface also showed that it is very stable, preserving its overall pattern for more than 300 h [5].

Many researchers have suggested diverse mechanisms to explain the nano-scale tribocharging phenomena. Depending upon one’s perspective, different aspects are highlighted. Some have explained the phenomena via electron transfer mechanisms [6,7] and others via ion transfer of confined water [8]. Material transfer is also regarded as a key mechanism [9]. The popular contact electrification perspective informs us that contacting a frictional surface affects the charging level, which can be easily amplified by the flexibility of PDMS [10,11,12]. Increasing the charging level may be possible via increasing interfacial frictions [13,14,15,16,17], but high-precision controls require an in-depth understanding of the underlying mechanisms and a delicate handling of the relevant physics. 

To find the physical mechanism of the nanopatterned tribocharge formation, we established theoretical models based on computational mechanics [3,4,18] and transparent machine learning [19]. The former revealed that the level of the nanoscale tribocharging depends strongly on the surface-tangential and surface-normal stresses experienced by the PDMS surface during the demolding step of the replica-molding process (Figure 1a). The latter unraveled the hidden rules behind the nanoscale tribocharging.

These analyses, however, were limited by their inherently qualitative and phenomenological nature. To facilitate the utilization of the nanopatterned tribocharging phenomenon, it is imperative to establish a more quantitative and analytical model. In this paper, we aim to establish multi-faceted physical and mechanical rationales that can explain the complex nanoscale tribocharge patterns on replica-molded soft-material surfaces. Specifically, we aim to apply the cohesive zone failure theory and the cumulative fracture energy model to our previous experimental results [4], perform quantitative analyses, and interpret the outcomes. The resulting theories shed more light on the root origin of the tribocharging phenomenon and widen its application potential. 

It is important to note the significance of this paper and its novelty compared to the prior works of the authors [3,4,5,18]. The previous works systematically document the experimental results, how to fabricate such intriguing nanoscale formations, and the summary of their behaviors from optical, electrical, and functional points of view. Therein, little has been done to elucidate the mechanical and analytical rationales behind the nanoscale tribocharging behaviors. In achieving better understanding and control of the nanoscale tribocharging phenomena in fabrication and application, this paper’s approach holds an important role for foundational knowledge.

## 2. Materials and Methods

### 2.1. Analysis Target Structures

For our analysis, we targeted the two nanoscale tribocharge patterns that we previously generated by replica molding PDMS with polymer molds with nanodome textures (Figure 1) [4]. Modeling the two tribocharge patterns side-by-side carries strong implications because the nanotextures on the two molds share the same nanodome geometry and diameter, differing only in their dome height and aspect ratios. The final tribocharge patterns, however, were distinctly different. As shown in Figure 1b,c schematically, replicating the mold with high- (and low)-aspect-ratio nanodome texture led to the formation of tribocharges in the shape of nanoscale rings (eclipses). Detailed AFM images and tribocharge distribution patterns, along with their overlap scans, are shown in Figure 2a–f. We analyzed the two phenomena using the cohesive zone failure theory and the cumulative fracture energy model, respectively.

### 2.2. Ring-Shaped Tribocharge Explained by the Cohesive Zone Failure Mechanism

From the perspective of mechanics, we first explain the rim concentration of the frictional stresses along the PDMS nanocup, which resulted in the ring-shaped tribocharge distribution pattern shown in Figure 2a. To that end, we hereby derived the equilibrium condition for an infinitesimal element on the surface of a PDMS nanocup. Figure 3 describes the infinitesimal interface element in the axisymmetric geometry; t stands for the thickness of the interface zone; p is the uniform vertical pressure caused by the peeling motion during the demolding process; *Q* is the internal shear force; σ and τ are the normal and frictional stresses on the interface, respectively. Each stress is assumed to be governed by the cohesive zone model (CZM)’s failure mechanisms [20].

By using the moment balance ∑r×F=0 and force balance ∑F=0 and by ignoring higher-order terms, we obtain the following: (1)$$\sigma \left(\theta \right)=p\left(1-\frac{t}{r}\right)\sin\theta ;\tau \left(\theta \right)=p\cos\theta$$

Figure 4 explains the stress states of the interface of the PDMS nanocup. In the shaded zone, which corresponds to a small θ (i.e., in the vicinity of the nanocup’s bottom rim), the debonding failure is governed primarily by the frictional stress. On the other hand, in the unshaded zone, with a greater θ, the debonding failure is governed mainly by the normal stress. Their influences switch the dominance at the intersection point of the surface-normal stress σ and the surface-tangential stress τ.

In essence, the CZM mechanism classifies the cohesive bonding failure into two scenarios (Figure 4). When the present shear stress (τ) exceeds the maximum shear stress limit of the interface ($\tau_{max}^{CZM}$), the cohesive bonding fails, while the surface-normal stress (σ) remains relatively small. The shaded regime in Figure 4 corresponds to this failure scenario. In this case, the shear stress can become very strong, and thereby, it can induce high-level tribocharging on the surface. In contrast, the second failure scenario can occur when the present normal stress exceeds the maximum normal stress limit (unshaded regime in Figure 4). In this scenario, the cohesive zone cannot develop sufficient shear stress, resulting in weak tribocharging.

The θ value of the intersection point, denoted as $\theta^{*}$, can be calculated by equating $\sigma \left(\theta^{*}\right)=\tau \left(\theta^{*}\right)$ as follows:(2)$$\theta^{*}=\mathrm{atan}\left[{\left(1-\frac{t}{r}\right)}^{-1}\right]$$

Calculating the exact thickness t of the debonding interface zone is beyond the scope of this paper. Still, we can glean important information by assuming t≪r, which leads to $\theta^{*}\approx \pi /4$. As shown in Figure 3, the apparent nanodome’s height (*H*) and radius (*R*) have the following geometric relationships: R=r⋅cosθ and H=r−r⋅sinθ. Thus, the aspect ratio, defined as *H*/*R*, can be given as follows: (3)$$Aspect\;Ratio\equiv \frac{H}{R}\equiv \frac{r-r\cdot sin\theta }{r\cdot cos\theta }.$$

We can obtain the critical aspect ratio of PDMS nanocup that can lead to rim-concentrated (i.e., ring-shaped) tribocharging:(4)$$Critical\;Aspect\;Ratio>{\left.\frac{1}{\cos\theta }-\sin\theta \;\right|}_{\theta^{*}}=\sqrt{2}-1\approx 0.414$$

In the high-aspect-ratio part of our previous work [4], the PDMS nanocup’s aspect ratio was ~0.62. This satisfies the critical aspect ratio condition, explaining the clear ring-shaped tribocharge distribution patterns observed in our previous work. 

If the aspect ratio is lower than the critical value, in lieu of frictional stress, the normal debonding stress will dominate the debonding failure, and the ring-shaped tribocharge pattern is less likely to be formed. In other words, this mechanics-based analysis becomes increasingly ineffective as the surface texture becomes flatter, decreasing its aspect ratio. In particular, the basic assumptions adopted for the derivations in this section start to break down in such geometries. Therefore, the next section will delve into the cumulative fracture energy to explain the eclipse-shaped tribocharge.

### 2.3. Eclipse-Shaped Tribocharge Explained by Cumulative Fracture Energy

To explain asymmetric tribocharging at low-aspect-ratio nanotextures (e.g., the tribocharge patterns in Figure 2d–f obtained from nanodomes with their aspect ratio at ~0.2), this section leverages another physical quantity, the *cumulative fracture energy*, which is more general than the stresses at equilibrium used in the previous section. We propose use of the tangential sliding distance $s_{t}$ (equivalent to *L_s_*) as a measure of the level of the contact electrification. Its impact was assessed through the evaluation and study of the cumulative tangential fracture energy: (5)$$G_{ft}\equiv \int \tau_{t}ds_{t}$$
where $\tau_{t}$ is the tangential traction. The value of $G_{ft}$ effectively describes the cumulative impact of the tangential stress and sliding during the debonding process and may elucidate the physical mechanism that generates the eclipse-shaped pattern shown in Figure 2d–f, which is very different from the ring-shaped pattern of the previous section, even though the replica molding was performed with similarly shaped nanodome textures.

For explanation, we adopted the mixed-mode bilinear cohesive zone model [20]:(6)$$\tau_{t}=K_{t}s_{t}(1-D_{m})$$
where $K_{t}$ is the cohesive stiffness, and $D_{m}$ is the damage parameter associated with the following: (7)$$\mathrm{Effective}\;\mathrm{Displacement}\;\mathrm{Jump}\;\equiv \sqrt{{\left(\frac{s_{n}}{s_{n}^{c}}\right)}^{2}+{\left(\frac{s_{t}}{s_{t}^{c}}\right)}^{2}}.$$

Here, $s_{n}$ is the interface-normal displacement, and $s_{n}^{c}$ is the value of $s_{n}$ at the completion of debonding. Similarly, $s_{t}$ corresponds to the interface-tangential sliding.

Our model geometry was a spherical cap surface, as shown in Figure 5a. Due to the spherical curvature of the interface surface, the level of tangential sliding differs at each interface point. Thus, we derived the projection of the global debonding displacement loading vector $u=(0,\delta ,∆{)}^{T}$ onto the tangential plane of the hemisphere, which is denoted as follows:(8)$$P_{u}=u-\left(u\cdot n\right)n$$
where $n\left(\varphi ,\theta \right)$ is the unit-normal vector of the hemisphere, $\varphi \in \left[0,\frac{\pi }{2}\right],and\;\theta \in \left[0,2\pi \right]$. For the debonding direction u, δ≪∆ is assumed since our actual debonding sequence starts from one side in the vertical direction. A typical r(φ,θ) represents the hemisphere and its derivatives as follows:(9)$$r_{\theta }=\partial r/\partial \theta ;\;r_{\varphi }=\partial r/\partial \varphi .$$

The outward positive normal unit vector is as given below:(10)$$n=-\frac{r_{\theta }\times r_{\varphi }}{\left\|r_{\theta }\times r_{\varphi }\right\|}={\left(\sin\varphi \cos\theta ,\sin\varphi \sin\theta ,\cos\varphi \right)}^{T}$$

Then, we can derive the magnitude of the projection: (11)$${\left\|P_{u}\right\|}^{2}=C^{2}\sin^{2}\varphi \cos^{2}\theta +{\left(\delta +C\sin\varphi \sin\theta \right)}^{2}+{\left(∆+C\cos\varphi \right)}^{2}$$
where
(12)$$C\left(\varphi ,\theta \right)=-(\delta \sin\varphi \sin\theta +\Delta \cos\varphi ).$$
${\left\|P_{u}\right\|}^{2}$ is equivalent to the tangential sliding distance $s_{t}^{2}$. 

Let $G_{ft,right}$ denote the cumulative tangential fracture energy of the half-spherical cap at θ∈[0,π], while $G_{ft,left}$ stands for that of the other half-spherical cap at θ∈[π,2π] (Figure 5a).
(13)$$G_{ft,right}\propto \int_{0}^{\bar{\varphi }}\int_{0}^{\pi }{\left\|P_{u}\right\|}^{2}d\theta d\varphi$$
(14)$$G_{ft,left}\propto \int_{0}^{\bar{\varphi }}\int_{\pi }^{2\pi }{\left\|P_{u}\right\|}^{2}d\theta d\varphi$$

Then, we can define a ratio that quantifies the difference in $G_{ft}$ in the two half-spherical caps as follows:(15)$$R_{G}(\bar{\varphi })\equiv {\frac{{|G}_{ft,left}-G_{ft,right}|}{{|G}_{ft,left}+G_{ft,right}|}}_{\bar{\varphi }\in [0,\pi /2]}$$

After some algebraic work, we can obtain that $\int_{\pi }^{2\pi }{\left\|P_{u}\right\|}^{2}d\theta =\pi \xi \left(\delta ,\;\Delta \right)+2\psi (\delta ,\Delta )$ and $\int_{0}^{\pi }{\left\|P_{u}\right\|}^{2}d\theta =\pi \xi \left(\delta ,\;\Delta \right)-2\psi (\delta ,\Delta )$, where the intermediate terms $\xi \left(\delta ,\;\Delta \right)\equiv \delta^{2}+\Delta^{2}\left(1-\cos^{2}\varphi \right)-{\frac{1}{2}\delta }^{2}\sin^{2}\varphi$, and ψ(δ,Δ)≡δΔsin⁡2φ. 

Thus, we can finally obtain the following:(16)$$R_{G}\left(\bar{\varphi }\right)=\frac{4∆\delta \sin^{2}\bar{\varphi }}{\;\;\;\frac{1}{4}\pi \left(6\delta^{2}\bar{\varphi }+4{∆}^{2}\bar{\varphi }+\left(\delta^{2}-2{∆}^{2}\right)\sin\left(2\bar{\varphi }\right)\right)\;\;\;}$$

The numerator corresponds to the energy difference ${|G}_{ft,left}-G_{ft,right}|$, while the denominator is associated with the total energy ${|G}_{ft,left}+G_{ft,right}|$ on the spherical cap ($\varphi \in [0,\bar{\varphi }]$). Numerical calculations of $R_{G}(\bar{\varphi })$ provide a clear picture regarding how the energy contrast varies as a function of the aspect ratio (H/R) that can be rewritten as follows:(17)$$H/R\equiv \left(r-r\cdot \cos\bar{\varphi }\right)/\left(r\cdot \sin\bar{\varphi }\right)=\tan\left(\bar{\varphi }/2\right)$$

As shown in Figure 5b, $R_{G}(\bar{\varphi })$ maximizes at the smallest aspect ratio at *H/R*~0.1. This explains the generation of the eclipse-shaped tribocharging pattern by near-flat spherical nanocups (Figure 2d–f). In contrast, as the aspect ratio increases, $R_{G}(\bar{\varphi })$ decreases rapidly. For a full hemisphere (i.e., *H*/*R* = 1.0), $R_{G}(\bar{\varphi })$ reaches its minimum, leading to a symmetric, ring-shaped tribocharging pattern (Figure 2a–c).

## 3. Discussion

### 3.1. Remarks on Tribocharging Patterns

We found that the new quantity $R_{G}\left(\bar{\varphi }\right)$, which is defined in Equation (16) as a measure of the difference in $G_{ft}$ in the two half-spherical caps, can give several important explanations to the observed tribocharging phenomena:

When the half nanocup is demolded in one direction and the other half in the opposite direction, the debonding displacement vector will be $u=(0,\delta ,∆{)}^{T}$ and $u=(0,-\delta ,∆{)}^{T}$ for right- and left-half caps, respectively. This leads to ${|G}_{ft,left}-G_{ft,right}|=0$, meaning that only a symmetric ring charge will be generated;A perfectly vertical debonding without any inclination at all (i.e., δ→0) also leads to ${|G}_{ft,left}-G_{ft,right}|\to 0$ as well as a ring charge for all aspect ratios (see the case δ/∆ = 0 in Figure 5b);When the debonding direction is controlled to have a large inclination angle, i.e., δ/∆≫0, $R_{G}(\bar{\varphi })$ rapidly increases, further distorting the symmetric tribocharging pattern.

Real-world experiments [4] agree well with these analytical-theory-based predictions. This model offers valuable insights into future extensions of the replica-molding-based nanopatterned tribocharging technique based on a wider variety of mold shapes, such as the pyramidal and cylindrical ones. 

Table 1 summarizes the authors’ previous experimental results that present the tribocharging phenomena with various geometries, all of which are in good agreement with the result of this proposed analytical investigation. As a special case of an eclipse tribocharging pattern, asymmetric charging on parallel nanoridges is also reported in Table 1. 

### 3.2. Computational Confirmation of the Ring-Shaped Tribocharging

We performed nonlinear debonding finite element analysis (FEM), which quantitatively measured the distributions of the maximum shear on the interface of PDMS spherical surface and the polymer mold to confirm the ring-shaped tribocharging, as shown in Section 2.1. Analytical challenges resulted from the spherically shaped interface and the severe material nonlinearities.

At the individual PDMS nanocup scale, the two surfaces of the interface are vertically displaced. However, the spherical interface makes the debonding occur in a “mixed” mode consisting of Mode-I (pure crack opening) and Mode-II (pure sliding). To reasonably describe the debonding process on the spherical interface, we adopted the mixed-mode CZM with both the material and geometric nonlinearities taken fully into consideration. 

The PDMS interface elements are modeled with its Young’s modulus *E* and the Poisson’s ratio v set to 1.8 MPa and 0.45, respectively, in accordance with the literature [21,22]. For the contact debonding regime, the CZM has its normal traction (denoted as $\sigma_{max}^{CZM})$ and maximum shear strength ($\tau_{max}^{CZM}$) set to $\sigma_{max}^{CZM}=\tau_{max}^{CZM}=$ 0.015 MPa. The CZM assumes that a complete separation occurs at 330 μm based on the relevant experiment results in the literature [23,24]. It should be noted that these strengths and separation limits are determined so that the debonding takes place clearly at the interface zone of the PDMS nanocup without causing any fracture to the PDMS fibrils. This choice was based on the observations that both the polymer mold and the PDMS nanocup consistently hold clear surfaces after repeated debonding attempts. All computational simulations were carried out on ANSYS [25].

Figure 6a summarizes the all-time maximum frictional stress along the circular path on the PDMS nanocup, starting from the left-bottom tip to the right tip. As observed from the tribocharging on the rim area of the PDMS nanocup, the maximum frictional stress is concentrated along the rim. Figure 6b–d show the sequential variation of the frictional stress on the outer surface of the PDMS nanocup during the debonding process. As shown in Figure 6, the PDMS nanocup’s lower rim region appears to have undergone higher frictional stresses in comparison to the top regions. This is associated with the fact that the “mixed” debonding mode is maximized in the inclined interface zones near the PDMS nanocup’s rim, while the flat or tangential interface areas are more likely suffer from a pure Mode-I or Mode-II, respectively.

### 3.3. Remarks on Limitations 

This paper focuses on the tribocharging phenomena occurring on the PDMS surface. The proposed analytical investigations may be applicable to other soft materials beside PDMS since the method mainly deals with mechanical relationship among geometric features, frictions, and the accumulated energy therein. However, due attention should be paid when the material becomes “too soft”, where the assumed geometric shapes are substantially altered due to large deformations during the detachment process. Future experimental research should consider additional material properties, such as the Young’s modulus and Poisson ratio, that can quantify the limits of the softness beyond, which the present analytical approach does not hold. For such general situations, more complicated approaches (e.g., [6,26]) should be incorporated into the present analytical methods. 

Another important question would be about whether the denser nanopatterns will affect/increase the tribocharging formation. This paper focused on individual nanopatterns and assumed each nanoscale tribocharging occurs independent of others. From the global point of view, the total tribocharging level will most likely increase when the nanopatterns take a denser formation. Friction is a key factor for the nanoscale tribocharging, and increasing shear friction can be accomplished by denser and taller nanopatterns. In a recent study [27], friction was found able to be increased by controlling the nanopattern’s geometry, and in fact, it can be designed and controlled. Connecting these recent findings to our approach will help researchers control the tribocharging of nanopatterned surfaces. 

## 4. Conclusions

This paper focused on an interesting phenomenon of nanopatterned tribocharge formation on the surface of elastomers. Although it has practical implications due to the ease of replica-mold-based fabrication, the physical rationale behind the phenomenon is little known. By separately applying two analytical approaches, i.e., the cohesive zone failure mechanism and cumulative fracture energy concept, this paper was able to analytically explain why the final tribocharge patterns became so different depending on the aspect ratio of the nanotextures on the mold, despite the similarities in the nanotextures’ morphology. We identified two regimes that generate symmetric (e.g., ring-shaped) and asymmetric (e.g., eclipse-shaped) tribocharging and theoretically found the criteria separating the two regimes. The results showed good agreement with our previous experimental observations. These models and analyses deepen our understanding of the triboelectrification phenomenon. This paper’s findings contribute to flexible wearable devices [28], high-density data storages [29,30], nanoxerography using electrical patterning of various types of particles [31], and so on. In particular, this paper elaborates the relationship between the geometric quantities (shapes, detachment directions, etc.) of the nanopatterns and the tribocharging level, which will help guide fabrication of advanced triboelectric devices. 

## Figures and Tables

**Figure 1 micromachines-15-00417-f001:**
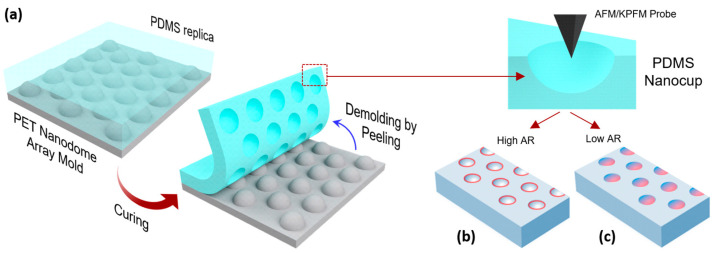
(**a**) The process of the replica-molding-based nanopatterned tribocharge formation. Depending on the aspect ratio (AR), (**b**) ring or (**c**) eclipse patterns were formed.

**Figure 2 micromachines-15-00417-f002:**
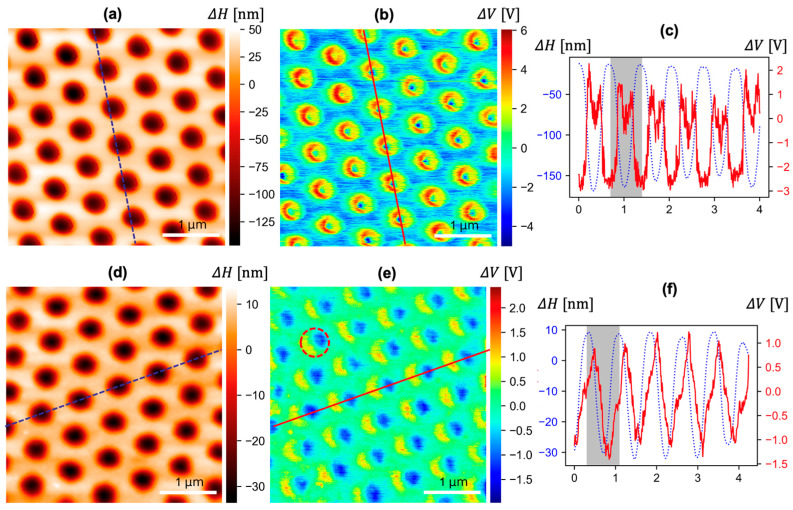
(**a**,**d**) AFM scans of the PDMS nanocup array’s surface topography (∆H), (**b**,**e**) KPFM scans of the surface potential (∆V) at the same spots, and (**c**,**f**) overlapped scans of ∆H and ∆V across the dotted and red straight lines. (**a**–**c**) The ring-shaped tribocharge distribution generated with the high-aspect-ratio (depth 157 ± 9 nm) nanodome mold and (**d**–**f**) the eclipse-shaped tribocharge distribution generated with the low-aspect-ratio (depth 48.7 ± 1.4 nm) nanodome mold. The radius of the nanodome was 250 nm in both cases. (Scale bars: 1 μm). Red dashed circle in (**e**) shows one nanocup with the exemplary asymmetric pattern, and the gray shadows in (**c**,**f**) correspond to one nanocup.

**Figure 3 micromachines-15-00417-f003:**
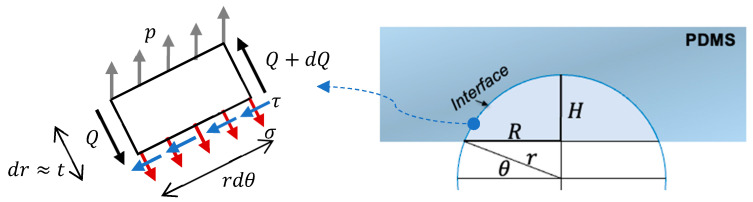
A free body diagram of an infinitesimal interface element of a PDMS nanocup. *R* and *H* are the apparent radius and height of the nanocup, whereas *r* is the radius of the reference sphere.

**Figure 4 micromachines-15-00417-f004:**
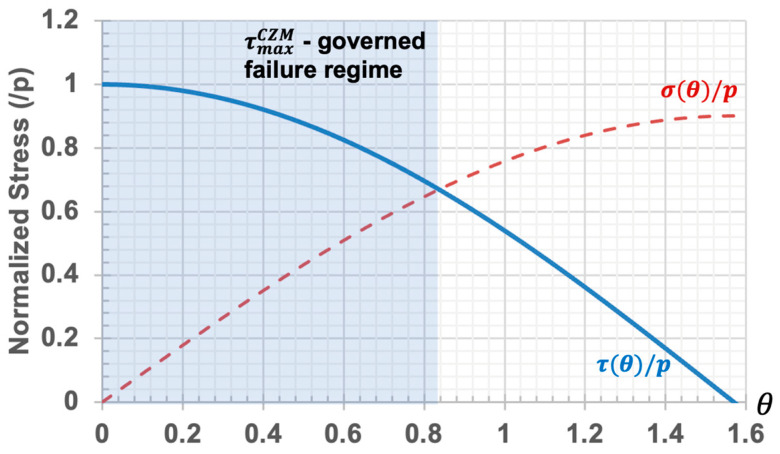
Frictional and normal stress plots of the interface on the PDMS nanocup. The shaded zone represents the regime where the frictional stress can reach the shear failure strength $\tau_{max}^{CZM}$ of the cohesive zone model.

**Figure 5 micromachines-15-00417-f005:**
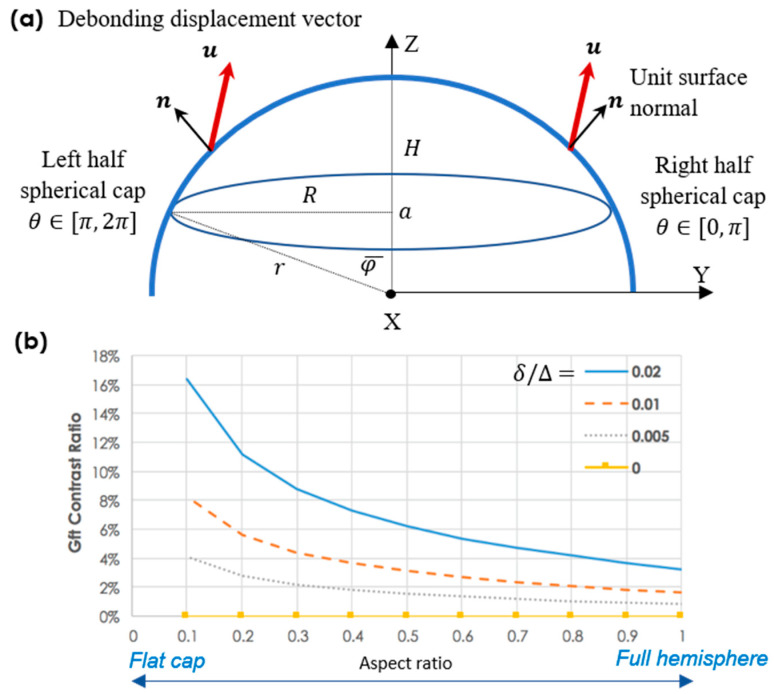
(**a**) Definitions of geometric terms used for calculation of the cumulative tangential fracture energy on the spherical cap. The *X* axis is normal to the plane, and debonding is assumed to have a slightly inclined (δ<∆) vertical direction in the YZ plane, (**b**) Variation of $R_{G}(\bar{\varphi })$ with varying aspect ratio *H/R* $\left(=1-\cos\bar{\varphi }\right)$ at each debonding direction. $\bar{\varphi }$ is the φ at the rim of the cap.

**Figure 6 micromachines-15-00417-f006:**
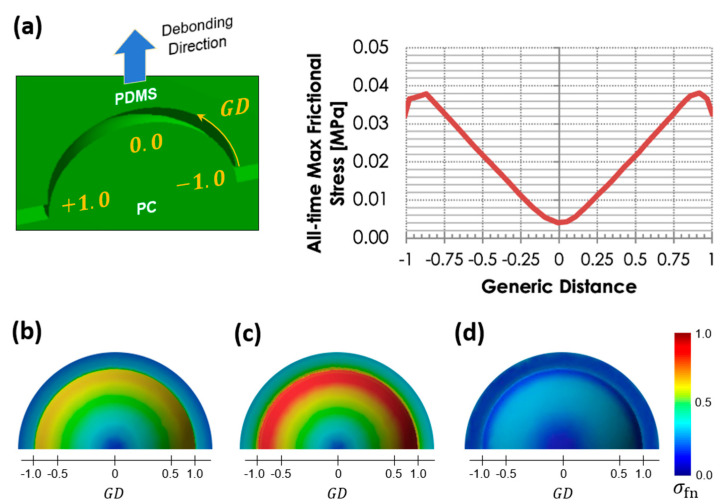
(**a**) Distribution of the all-time maximum frictional stress along the circular path of the PDMS nanocup during the debonding process. Generic distance (GD) ranges from −1 to 1, meaning the left to the right tip at the bottom of the PDMS nanocup. (**b**–**d**) Bottom view of the variation of frictional stress (MPa) on the outer surface of the PDMS nanocup along the incremental progress of the debonding process with (**b**) initial, (**c**) right before debonding, and (**d**) post-debonding stages. A PDMS nanocup with the aspect ratio 0.6 demonstrates ring-shaped tribocharging. Nonlinear FEA with the cohesive zone failure mechanism was adopted.

**Table 1 micromachines-15-00417-t001:** Summary of different nano-tribocharging patterns that are in good agreement with this paper’s analytical findings. Figure numbers correspond to the authors’ reference. Due to the copyright issue, direct snapshots of those figures are not included.

Nanoscale Tribocharging Patterns	Figures	Reference
Ring shape	Figure 2	[3]
Ring shape	Figure 2g	[2]
Eclipse shape	Figure 4c	[19]
Eclipse shape	Figure 2i	[2]
Asymmetric on parallel nanoridges (special case of eclipse shape)	Figure 5a	[19]
Asymmetric on parallel nanoridges (special case of eclipse shape)	Figure S2	[2]

## Data Availability

All data are available upon request to the corresponding authors.

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
