# Peer review of "Analytical Investigation of Replica-Molding-Enabled Nanopatterned Tribocharging Process on Soft-Material Surfaces"

_micromachines, 2024, doi:10.3390/mi15030417_

Round 1

Reviewer 1 Report

Comments and Suggestions for Authors

Author Response

Please see the response document.

Reviewer 2 Report

Comments and Suggestions for Authors

The paper "Analytical Investigation of Replica Molding-enabled Nanopatterned Tribocharging Process on Soft Material Surfaces" delves into the generation of nanopatterned tribocharge on elastomer surfaces using nanotextured molds. It builds upon the authors' previous experimental findings by introducing analytical models to explain the observed phenomena, focusing on cohesive zone failure and cumulative fracture energy. This work aims to provide a deeper understanding of the mechanisms behind nanopatterned tribocharge formation, contributing significantly to the field of nanotechnology and materials science. I have a few comments as following.

1. The manuscript touches upon less novelty of the approach compared to previous studies, including the author’s own work which I believe is very close to the present work:

@Replica molding-based nanopatterning of tribocharge on elastomer with application to electrohydrodynamic nanolithography (https://doi.org/10.1038/s41467-018-03319-4),

@Nanoscale Modulation of Friction and Triboelectrification via Surface Nanotexturing (10.1021/acs.nanolett.8b04038)

@Mechano-Triboelectric Analysis of Surface Charge Generation on Replica-Molded Elastomeric Nanodomes (https://doi.org/10.3390%2Fmi12121460)

Therefore, a more thorough comparison need to be addressed by highlighting the advancements made in this study, such as improved model accuracy or new insights into the tribocharging process, would provide clearer value to the reader.

2. The use of cohesive zone failure and cumulative fracture energy models is well-justified and effectively applied. However, it would be beneficial to include more detailed comparisons with experimental data to validate the models further.

Author Response

Please see the response document. 
